# Effect of Ceramic Formation on the Emission of Eu^3+^ and Nd^3+^ Ions in Double Perovskites

**DOI:** 10.3390/ma14205996

**Published:** 2021-10-12

**Authors:** Natalia Miniajluk-Gaweł, Bartosz Bondzior, Karol Lemański, Thi Hong Quan Vu, Dagmara Stefańska, Remy Boulesteix, Przemysław Jacek Dereń

**Affiliations:** 1Institute of Low Temperature and Structure Research, Polish Academy of Science, Okolna2, 50-422 Wroclaw, Poland; b.bondzior@intibs.pl (B.B.); k.lemanski@intibs.pl (K.L.); q.vu@intibs.pl (T.H.Q.V.); d.stefanska@intibs.pl (D.S.); p.deren@intibs.pl (P.J.D.); 2Institute of Research for Ceramics (IRCER), UMR CNRS 7315, University of Limoges, F-87068 Limoges, France; remy.boulesteix@unilim.fr

**Keywords:** double perovskite, rare earth ions, ceramic materials, SPS method, scintillator

## Abstract

Herein, the structure, morphology, as well as optical properties of the powder and ceramic samples of Ba_2_MgWO_6_ are presented. Powder samples were obtained by high temperature solid-state reaction, while, for the ceramics, the SPS technique under 50-MPa pressure was applied. The morphology of the investigated samples showed some agglomeration and grains with a submicron size of 490–492 µm. The theoretical density and relative density of ceramics were calculated using the Archimedes method. The influence of sample preparation on the position, shape, and character of the host, as well as dopants emission was investigated. Sample sintering enhances regular emission of WO_6_ groups causing a blue shift of Ba_2_MgWO_6_ emission. Nonetheless, under X-ray excitation, only the green emission of inversion WO_6_ group was detected. For the ceramic doped with Eu^3+^ ions, the emission of both host and dopant was detected. However, for the powder efficient host to activator energy, the transfer process occurred, and only the magnetic dipole emission of Eu^3+^ was detected. The intensity of Nd^3+^ ions of Ba_2_MgWO_6_ powder sample is five times higher than for the ceramic. The sintering process reduces inversion defects and creates a highly symmetrical site of neodymium ions. The emission of Ba_2_MgWO_6_:Nd^3+^ consists of transitions from the ^4^F_3/2_ excited level to the ^4^I_J_ multiplet states with the dominance of the ^4^F_3/2_→^4^I_11/2_ one. The spectroscopic quality parameter and branching ratio of Nd^3+^ emission are presented.

## 1. Introduction

Double perovskites materials, i.e., A_2_BB’O_6_, have recently been receiving more and more attention from scientists due to their fascinating physical and chemical properties stemming from their flexible structural modification ability. As a result, double perovskites found a wide range of applications in white light-emitting phosphors [1,2], transparent and translucent ceramics [3,4,5,6], dielectric materials [4,7,8], optical thermometers [9,10,11], scintillators [12], etc. Among them, scintillators are widely used in medical diagnosis, computed tomography, non-destructive inspections, security, nuclear radiation detection, etc. Scintillators are known as radiosensitive luminescent materials that can convert high-energy photons (X-rays or γ-rays) to low-energy photons or electrical charges [13]. Some conventional scintillators, such as Lu_3_Al_5_O_12_:Ce^3+^, require very high temperature sintering and long annealing time [14,15], while modern scintillating materials (mostly lead halide elpasolite perovskites structure $A_{2}^{+1}B^{+1}C^{+3}X_{6}$) exhibit low X-ray light yield, instability, and toxicity [12,13,16]. Therefore, finding novel scintillating materials, which require low-dose irradiation, high stability, and high spatial resolution, and are lead-free, has become extremely urgent.

Tungstate double perovskite is a kind of environmental friendly material comparable with fluorides, sulfides, and silicates from the synthesis point of view [17,18]. Ba_2_MgWO_6_ (BMW), a representative of tungstate double perovskites, has demonstrated a great potential for making ceramics. There are a few studies related to this structure. The first work is dedicated to the synthesis and preparation of BMW ceramics in 2004 [3], which showed the possibility of obtaining a ceramic from BMW for planar RTD applications. Two other publications were focused on the microstructure [5] and microwave dielectric characteristics [4] of BMW ceramics. However, the ceramics in the abovementioned studies were only prepared by traditional heating. The last one, published recently, is devoted to BMW:Ce^3+^ used in an advanced sintering technique, namely the spark-plasma-sintering method (SPS) as a ceramic-based potential scintillator [6].

In the last few decades, polycrystalline materials, including transparent ceramics, witnessed rapid growth due to considerable needs in numerous fields. Ceramics are far superior to many other materials in terms of resistance to rapid temperature changes, and they can work in harsh conditions, such as corrosive or chemical environments, high temperature or high pressure, etc. In ceramics, there will also be dispersion phenomena; however, they will be significantly smaller than in other forms of materials.

SPS is known as a densification process that utilizes a combination of high pressure and temperature resulting from a high current flow in a conductive die. The main advantage of this method is that it allows a rapid heating rate up to 200–300 °C/min; therefore, the waiting time for elevating the temperature is shortened. Recently, transparent ceramics sintered by SPS method have been extensively investigated [6,19,20,21,22,23]. The temperature and pressure parameters play an important role in the quality of ceramics, especially transparency. To obtain a highly transparent ceramic, one should increase sintering time and adjust pressure so as to improve the density of sintered samples [6]. Besides, the porosity, known as the light scattering center, is also a critical issue. The higher the porosity, the lower the transparency of the ceramic. The latter can be influenced by many factors, such as vacancies, color centers, pores, or impurities phases. Thus, to obtain a ceramic with good transparency properties, it is absolutely necessary to examine the polycrystalline powder’s characteristics, including phase purity, grain size, morphology, as well as its sintering ability.

The aims of this study are to verify the possibility of ceramic preparation based on BMW polycrystalline powders and investigate their characteristics as a scintillator. In this study, BMW (undoped and doped with Eu^3+^ and Nd^3+^) were synthesized by the conventional solid-state method. A comprehensive understanding of ceramic fabrication from polycrystalline powders of both host and doped samples is presented. To verify the applicability of polycrystalline powders for ceramic preparation, the structure analysis was conducted and the thermal behavior of the samples was examined. The qualified samples were used to prepare ceramics by employing the SPS technique and their microstructural characteristics were investigated. Furthermore, investigation into spectroscopic properties under X-ray irradiation of undoped and doped ceramics was performed. The results of both BMW ceramics showed that their spectra were located in a visible region after X-rays excitation, which offers a novel approach of this material as a scintillator.

## 2. Experimental

### 2.1. Samples Preparation

Ba_2_MgWO_6_ (BMW), Ba_2_Mg_0.99_Eu_0.01_WO_6_ (BMW-Eu), and Ba_2_Mg_0.99_Nd_0.01_WO_6_ (BMW-Nd) phosphors were prepared by the conventional high-temperature solid-state reaction method. The following chemical reagents were used in the synthesis: Ba(CH_3_COO)_2_ (Alfa Aesar, 99%); Mg(CH_3_COO)_2_·4H_2_O (Merck, 99.5%); WO_3_ (Alfa Aesar, 99.998%); Li_2_CO_3_ (Alfa Aesar, 99.998%); and the source of dopant ions, i.e., Eu^3+^ as Eu_2_O_3_ (Alfa Aesar, 99.9999%) and Nd^3+^ as Nd_2_O_3_ (Alfa Aesar, 99.99%). The stoichiometric amounts of the starting materials with 10% excess of Mg(CH_3_COO_2_)·4H_2_O, to suppress the formation of BaWO_4_ as an impurity, were first mixed and ground thoroughly in a mortar for 20 min. Substitution of a divalent ion (Mg^2+^) with a third oxidation state lanthanide ion (Eu^3+^ and Nd^3+^) can cause defects in the matrix, which occur to compensate the created charges locally in the host. For this reason, charge compensation was applied through the addition of lithium in an amount of 50%, based on the amount of dopant ions. The samples were pre-sintered at 600 °C/6 h and re-sintered at 1300 °C/6 h two times in air atmosphere in a muffle furnace in an Al_2_O_3_ crucible. The products were obtained after cooling the samples to room temperature naturally and subsequently they were ground into fine powder.

The obtained powder was then sintered using the spark plasma sintering (SPS) method (SPS Dr Sinter 825, Syntex, Kawasaki, Japan). First, 1.3 g of powder was introduced in a graphite die with a 10-mm inner diameter. Next, the samples were sintered under vacuum at temperatures of 1350 °C, under uniaxial pressure of 50 MPa. Both the heating rate and the cooling rate were set to 100 °C/min. Figure 1 presents the diagram of the SPS sintering thermomechanical schedule (RT = room temperature). After sintering, the samples were annealed at 1000 °C for 10 h in air for decoloring (reoxidation) and graphite burnout. The last stage of the process was machining and polishing.

### 2.2. Research Techniques

The crystalline phases were identified and indexed by the X-ray diffraction (XRD) analysis (D8 ADVANCE, Bruker, Karlsruhe, Germany) using monochromatic Cu K_ɑ1_ radiation (λ = 1.54056 Ǻ). The data were collected in a 2θ range from 10° to 90° with a step of 0.026.

The microstructure of the sample was investigated by a scanning electron microscope FEG-SEM (Quanta 450, FEI, Thermo Fisher Scientific, Waltam, MA, USA).

The apparent density of materials was determined using the Archimedes method in absolute ethanol (density δeth = 0.79 g/cm^3^), and then the relative density was obtained using a theoretical density of BMW double perovskite δBMW = 7.20 g/cm^3^. The measurement error of this technique is of about ±1% in our case as the sample mass is low. As part of the density measurement by the Archimedes method, the following masses were determined: *m*1—the mass of dry pellet; *m*2—the mass of pellet immersed in ethanol; and *m*3—the mass of pellet impregnated with ethanol. The apparent density, open porosity, and relative density of the samples before and after sintering were calculated using the following Equations (1)–(3):
(1)$$\delta_{app}=\frac{m1}{m3-m2}\;\times \;\delta_{\mathrm{eth}}$$
(2)$$\rho =\frac{\delta_{\mathrm{app}}}{\delta_{\mathrm{BMW}}}\;\times \;100$$
(3)$$\Pi_{o}=\frac{m3-m1}{m3-m2}\;\times \;100$$
where δ_app_ is the apparent density (i.e., with closed porosity), Π_o_ is the open porosity (in %), and ρ is the relative density (in %).

As an excitation source, a 266-nm line from a Nd:YAG pulse laser and a laser diode operating at 808 nm were used. The emission spectra were recorded using a Hamamatsu PMA-12 detector (Hamamatsu, Japan). An InGaAs detector equipped with VUV McPherson spectrometer (Chelmsford, MA, USA) was used for NIR analysis. Decay profiles were obtained on a Lecroy digital oscilloscope (Chestnut Ridge, NY, USA). A Varian VF-50J/S RTG tube (Salt Lake, UT, USA) was used as an X-ray radiation source. The voltage and amperage for the X-ray source were 40 kV and 0.7 mA, respectively.

To ensure repetition and comparatives of the results, a special setup for measuring the emission in the same condition was applied when it comes to the excitation source and its power, the detector, integration time, geometry, etc. As for the illuminated material, the powder was placed in a flat holder with a quartz window with negligible absorbance, while the ceramic sample was measured without a holder in the same plane as the powder. The spot of the illuminating source was small enough so that the illuminated area of the powder and the ceramic was the same.

Structural and microstructural properties were tested at the University of Limoges, while spectroscopic measurements were made at the Institute of Low Temperature and Structure Research Polish Academy of Sciences.

## 3. Results and Discussion

### 3.1. BMW Powders Characterizations

Initially, structural studies of polycrystalline powders were conducted in order to verify whether the material has the ability to be compacted and thus form ceramic materials. For this purpose, a structure analysis by X-ray diffraction and SEM images were performed.

Figure 2 shows X-ray diffraction patterns of polycrystalline powders. It was observed that the diffraction peaks of the samples match well with the ICDD card no. 70-2023 for BMW. Each of the materials has a cubic double perovskite structure, with the space group *Fm-3m* and the following crystallographic structural parameters: lattice parameter a=8.1120 Å, unit cell volume $V=533.81{\;Å}^{3}$, Z=4, and theoretical density δth = 7.2. Our current interest is focused mainly on obtaining the pure composition. Transparent ceramics would be obtained only for the pure host, because the secondary phases of the different refractive index will act as light scattering centers. In the case of the tested materials, the XRD results confirm that all polycrystalline powders obtained the structure of a double perovskite, and also the Eu^3+^ and Nd^3+^ ions were introduced into the crystal lattice with no change of their structure. However, only BMW-Eu has a single phase, while BMW and BMW-Nd have the main crystalline phase of Ba_2_MgWO_6_, accompanied by small amounts of BaWO_4_ (Figure 2).

The SEM images of the BMW, BMW-Eu, and BMW-Nd powders are presented in Figure 3. The grains tend to agglomerate, but individual grains also appear. The grains are of relatively large sizes in the range of 490 nm–2 µm; however, the BMW-Nd composition has the smallest size (Table 1). The analysis of SEM images shows that the solid-state synthesis produces isotropic, slightly agglomerated, and submicrometric double perovskite powders, such as Ba_2_MgWO_6_; therefore, these materials should exhibit good sinterability. 

In our previous work [6], an analysis of the thermal behavior with the natural sintering method was performed using dilatometry and the TG-DSC analysis in air, which supplements the test results obtained. The measurements were caried out on the BMW host and were described in detail in the previous paper [6], and therefore were not included in this paper. Overall, the dilatometry results show that our BMW powders have excellent compaction ability, which may be related to the low friction and rearrangement forces experienced during compaction. In addition, the sinterability of BMW synthetic powders remains low during natural sintering, which can be explained by the large grain size. The obtained results of TG-DSC indicate that the sintering temperature of this material should not exceed 1400 °C. For this reason, the SPS sintering method has been studied to help lower the densification temperature. In conclusion, Ba_2_MgWO_6_ powder with a double perovskite structure can be used to obtain ceramics.

### 3.2. Study of BMW Sintering by SPS

BMW powders were sintered by SPS according to the schedule presented in Figure 1. Figure 4 shows the sintering profiles of double Ba_2_MgWO_6_ perovskites. The upper profiles show the shrinkage (∆L, mm) of the material as a function of temperature, which tells us if the material has been compacted and if a ceramic material has been formed. However, the lower profiles show the change in pressure (p, kN) supplied to the system, also depending on the temperature. The sintering profiles are difficult to compare with each other because during the process there was a problem with the applied pressure was too small, which led to poor pressure control. Nevertheless, on the basis of the obtained results, it can be concluded that all samples have very similar compaction behavior. The first shrinkage due to rearrangement of the powder particles is observed at about 600 °C when the starting pressure is applied. The second contraction is observed due to compaction starting in the range of about 1150–1200 °C. In this temperature range, a shrinkage of about 1 mm is observed in all cases. The corresponding relative shrinkage is approximately 26%, calculated on the basis of Equation (4):(4)ΔL(T)[00]=100× L0¯ΔL(T)[mm]
where ∆L(T) [%] is the shrinkage, ∆L(T) [mm] is the thickness variation (ΔL(T) = L(T) − L_0_), and Lo is the initial thickness equal to around 3.8 mm.

### 3.3. Microstructural Features of BMW Ceramics

Ceramic materials obtained by the SPS method on the basis of BMW, BMW-Eu, and BMW-Nd polycrystalline powders were microstructurally analyzed. For this purpose, XRD, SEM, and density measurement by the Archimedes method were performed. Overall, the obtained results confirm the obtaining of well-dense double perovskite ceramics. Table 1 summarizes the density parameters, determined by the Archimedes method. All samples appear to be almost densified—relative density is more than 84% (δ > 84%).

The structure of the double perovskite Ba_2_MgWO_6_ of the obtained ceramic materials, sintered at 1350 °C by the SPS method, was also confirmed. It can be observed in Figure 5 that the diffraction peaks of the samples match well with the ICDD card no. 70-2023 for cubic Ba_2_MgWO_6_. The materials are practically single-phase, only in the case of BMW-Eu material, whereby there is an additional phase of BaWO_4_ with the ICDD card no. 43-0646 (Figure 5). By comparing the X-ray diffraction patterns of polycrystalline powders and ceramic materials (Figure 2 and Figure 5), it can be noticed that the sintering process does not change the structure of the obtained materials, but affects their phase nature. In the case of powders, only BMW-Eu is single-phase, while BMW and BMW-Nd also have small amounts of BaWO_4_ (Figure 2), while for ceramics, the materials are practically single-phase, except BMW-Eu materials (Figure 5). One may even be tempted to say that the SPS sintering method helps to obtain materials with higher phase purity (Figure 2 and Figure 5).

The SEM micrographs of the BMW ceramics are shown in Figure 6. Dense ceramic materials with high relative density (higher than 84%, see Table 1) were obtained for each composition. These results show that the SPS schedule at 1350 °C under 50 MPa of uniaxial pressure seems to be well adapted to densify such ceramic materials. From the SEM photos, it can be seen that all the materials subjected to the SPS sintering tests were compacted. The photos show grains connected by distinct shapes with a few visible pores. The SEM images confirm that all materials were sintered with limited grain growth, which allows good microstructure homogeneity and fine microstructure to be maintained. The average grain size of the polycrystalline powders and the obtained ceramic sinters did not differ significantly from one another (see Table 1). This shows that the application of SPS to submicrometric powders allowed to limit the grain growth due to the fast compaction. On the basis of the previously described X-ray diffraction patterns, the obtained ceramic materials (Figure 5) are practically single-phase, except for in the case of the BMW-Eu material where there is an additional phase of BaWO_4_. The dark spaces visible in the SEM images are pores, while the impurities of the structure in the form of an additional phase BaWO_4_ are visible in the images in the form of heterogeneity, i.e., where bright and darker areas in the form of streaks are visible. However, it should be emphasized that the additional phase in the form of BaWO_4_ is much more visible on the X-ray diffraction pattern in the form of an additional reflex; the SEM images only confirm this.

### 3.4. Spectroscopic Properties of BMW Double Perovskites

The maximum of BMW powder emission is at 522 nm and the emission spectrum (see Figure 7) consists of two bands assigned to the blue-green emission of regular (in the range 350–500 nm) and yellow-green emission (in the range 450–800 nm) of inverse WO_6_ groups. The maximum of the ceramic emission spectrum is blue shifted with the maximum at 503 nm and the proportion of regular/inverse emission is larger. The overall emission intensity is about nine times stronger for ceramics, but the relative intensity of the inverse WO_6_ group is weaker, probably due to the fact that the sintering under big pressure does not allow for the formation of inversion defects in ceramics.

The BMW pure ceramics sample was excited also using the X-ray radiation (see Figure 8a). The emission from the WO_6_ groups was observed. However, this was mainly at the green region, suggesting that the inverse WO_6_ groups are more sensitive to this high energy radiation than the regular groups. The observations of the visible light under X-rays lead to creating new opportunities of using this material as a scintillator to detect high-energy ionizing radiation. The emission intensity of regular and inverse WO_6_ groups not only strongly depends on the form of the sample but also on the source of the excitation. As a consequence, the emission chromaticity CIE (*x*, *y*) of investigated samples change from (0.222, 0.329) by (0.309, 0.422) to (0.354, 0.486) for ceramics, powder, and ceramics under X-ray excitation, respectively (Figure 8b). The color of observed emission varies from bluish-green for ceramics to yellowish-green for powder and yellow-green for ceramic under X-ray excitation.

Contrary to the host emission, the Eu^3+^ emission (see Figure 9) seems to be little affected by the sintering process. The emission spectrum of the Eu^3+^ ions conserves its characteristics for both powder and ceramics, with dominant ^5^D_0_ → ^7^F_1_ magnetic–dipole transition around 596 nm and weak electric–dipole transitions ^5^D_0_ → ^7^F_J_, where J = 0,2,3,4,5,6 characteristic of the highly symmetrical position of the Eu^3+^. The regular/inverse WO_6_ groups emission ratio (visible in the inset of Figure 9) is visibly lower, which confirms the energy transfer between the regular WO_6_ and Eu^3+^ ions, previously reported in [24]. The emission of the tungsten group is not visible for powder materials (see Figure 9).

Bode and Oosterhout [25] assigned blue-green and yellow-green emission bands to the regular and irregular emission of the WO_6_ group, respectively. In the latter, the tungsten ions, instead of entering their permanent place at BMW, replace the Mg^2+^ position. It has been shown [24] that, with the increase in Eu^3+^, the emission intensity of WO_6_ decreases, which confirms that the excitation energy is transferred from the host to the dopant. Based on Dexter’s theory, Blasse analyzed the energy transfer from WO_6_ groups to Eu^3+^ ions [26]. He argues that the probability of energy transfer through the exchange mechanism is equal to the product of the overlap of the orbital wave functions W^6+^ and Eu^3+^, and that the overlap of the emission energy of the WO_6_ groups with the states of absorption of Eu^3+^ ions. The orbital overlap, and hence the energy transfer mechanism, is regulated by the angle between the W^6+^–O^2-^–Eu^3+^ bonds. If the angle is 90°, the orbital overlap is negligible, and the largest is observed for the angle of 180° [9,24]. In the previous report on BMW, Eu^3+^ prepared by the mechanical–chemical method [9], it was calculated that the angle between W^6+^ (_regular_)–O^2-^ Eu^3+^ is equal to 180º; however, the angle between W^6+^ (_irregular_)–O^2-^–Eu^3+^ is equal to 135°. This is why energy transfer to Eu^3+^ ions mainly occurs from regular WO_6_ groups.

To evaluate the impact of the form of material on the efficiency of the energy transfer process from the WO_6_ groups to Eu^3+^ ions, the energy transfer efficiency $\left(\eta_{\mathrm{ET}}\right)\;$ was calculated using the following Equation (5).
(5)$$\eta_{\mathrm{ET}}=1-\frac{I_{\mathrm{Eu}3+}}{I_{o}}$$
where $I_{{\mathrm{Eu}}^{3+}}$, $I_{o}$ is the integrated intensity of the sample with Eu^3+^ and the host, respectively. The energy transfer efficiency between WO_6_ groups and Eu^3+^ ions was found to be 93.3% for the ceramic as compared to 99.2% for the sample in a powder form. The sintering process, while enhancing the host emission intensity, slightly restricts the energy transfer to Eu^3+^ ions.

The luminescence lifetimes of the main Eu^3+^ ions emission were calculated using the integral equation due to the non-exponential character of the decay curves. The results are given in Figure 10. The shorter lifetime of the Eu^3+^ emission in the BMW ceramic material is a result of the deformation of the vicinity of the Eu^3+^ ions due to a large pressure applied to the powdered materials during sintering. Even a slight distortion of the environment will lower the ideal O_h_ symmetry with the inversion center in which the Eu^3+^ ions are located, making the observed emission more likely, and therefore shortening the lifetime of the excited ^5^D_0_ level.

The ratio of the integrated intensities between ^5^D_0_→^7^F_2_ (electric dipole) and ^5^D_0_→^7^F_1_ (magnetic dipole) transitions of the Eu^3+^ ions is so-called the asymmetry factor R, and can be calculated with the Equation (6):(6)R=∫I(D05→F27)∫I(D05→F17)

This ratio describes the asymmetry around the Eu^3+^ ions. The obtained values are the same for the BMW-Eu^3+^ ceramics and powders, and are equal to 0.44. This indicates the high symmetric environment around the europium (III) ions in the investigated samples.

The emission spectra of Nd^3+^ in powder and ceramic BMW are presented in the normalized form in Figure 11. The emission intensity is reduced for the ceramics sample by the factor of 5–6. This may suggest that the sintering process either eliminates the residual foreign phases present in the powder or, similarly to the results of the pure host, reduces the number of inversion defects and creates a highly symmetrical surrounding for Nd^3+^ ions.

The ^4^F_3/2_ → ^4^I_11/2_ transition is the strongest for all samples. At 77 K, the shape of the peaks is altered. Interestingly, the rarely observed ^4^F_3/2_ → ^4^I_15/2_ occurs in all cases; however, it is relatively stronger for the BMW ceramics, especially at the liquid nitrogen temperature.

The spectroscopic quality parameter X_Nd_, defined as a ratio of Judd–Ofelt intensity parameters, was calculated with using the formula described by A. Kaminskii [27].
XNd (F3/24) =Ω4Ω6=0.765 YNd−2.96
where Y_Nd_ = I_J, 11/2_/I_J, 13/2_ and I_J_ is integrated emission intensity; J is the total angular momentum quantum number.

Depending on the sample, X_Nd_ has a different value in the range from about 0.65 up to 2.81 (see Table 2). At a lower temperature, when the influence of phonons is lower, both ceramics and powder of BMW:Nd^3+^ had a higher value of the SQP.

All samples were measured both at 77 K and room temperature. The reason the results were not presented in their entirety is the following: only Nd samples exhibit a relevant difference in the emission spectra measured at both temperatures. The emission spectra of other samples are virtually identical in both temperatures.

Based on the emission spectra, the branching ratio of emission from ^4^F_3/2_ to ^4^I_J_ multiplets was obtained (see Figure 12). The contribution of the emission, depending on the sample and measurement temperature, shows low differences for the investigated samples at different temperatures.

The differences in luminescent properties between powdered and SPSed samples are mainly due to three reasons.

The first has already been mentioned in the description of the lifetimes of the emissions. The high pressure applied during the sintering of ceramics causes slight changes in the symmetry of the environment of Eu^3+^ ions. Even slight changes in the environment reduce the ideal symmetry of the ions observed for the powder material. In the lowered symmetry, the participation of the electric dipole in the transitions increases, and, therefore, the ion emission transitions, either Eu^3+^ or Nd^3+^, become more admissible because the selection rules are partially released. This, in turn, results in a shorter lifetime of the excited level and a higher emission intensity.

The second reason for the differences in spectroscopic properties of the BMW powder and ceramic materials is the influence of the pressure used in sintering ceramics on the formation of more WO_6_ inversion sites. This can most prominently be seen in Figure 9, which shows a much larger share of the emission of irregular groups of WO_6_, while, in the powder material, this emission is almost absent (see inset in Figure 9). This observation is also confirmed by the approximately 6% reduction in the efficiency of transfer of emissions from the host to the dopant for ceramics. As mentioned above, the irregular groups, i.e., WO_6_, hardly transfer energy to Eu^3+^. 

Finally, the third effect of changes in the emission properties of ceramic materials results from the reduction in surface area and grain boundaries that occur in powder samples. This reduction primarily affects emission intensity by eliminating light scattering at grain boundaries and by improving overall light output.

## 4. Conclusions

The conventional solid-state reaction and the SPS technique were successfully applied to prepare powder samples and ceramics of Ba_2_MgWO_6_. All of them, pure BMW, BMW: 1% Eu^3+^, and BMW: 1% Nd^3+^, possess a regular double perovskite structure with the space group *Fm-3m*. SEM images showed that the prepared ceramics were well compacted; however, some pores were also visible. The highest relative density δ determined by the Archimedes method equals 91.99% for the undoped BMW. The proportion between regular and irregular emission of WO_6_ groups depends on sample forms and the excitation source. As was shown, regular WO_6_ emission was enhanced for the BMW ceramic in contrast to BMW powder. However, under X-ray excitation, only irregular emission of WO_6_ groups was detected. Green emission located at 522 nm after X-ray excitation offers a novel approach to this material as a scintillator. The sample also influences the energy transfer WO_6_ → Eu^3+^ efficiency; for powder $\eta_{\mathrm{ET}}$, it is almost 100%, while for the ceramic, $\eta_{\mathrm{ET}}=93.3\%$. As a consequence, both emissions of host and dopant ions are presented for the ceramic. High temperature and pressure during SPS reduce the inversion defect; as a result, characteristic Nd^3+^ emission positioned in a high symmetry site was detected with the dominance of ^4^F_3/2_ → ^4^I_11/2_ transition.

## Figures and Tables

**Figure 1 materials-14-05996-f001:**
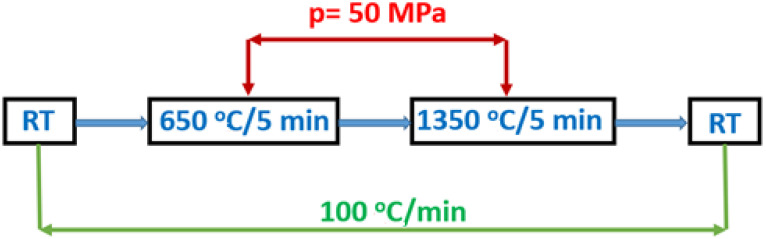
Diagram of SPS sintering thermomechanical schedule.

**Figure 2 materials-14-05996-f002:**
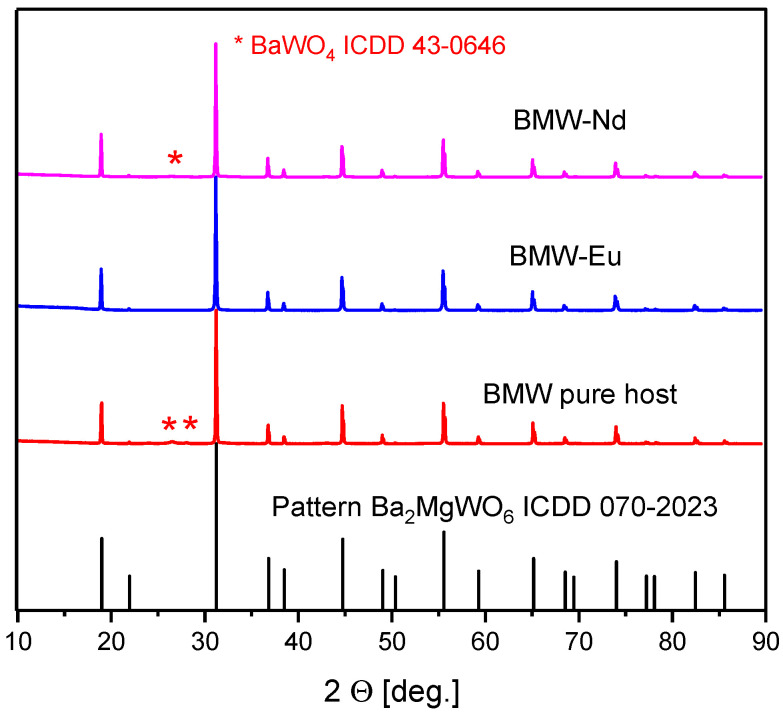
XRD patterns of the polycrystalline powders: BMW, BMW-Eu, and BMW-Nd.

**Figure 3 materials-14-05996-f003:**
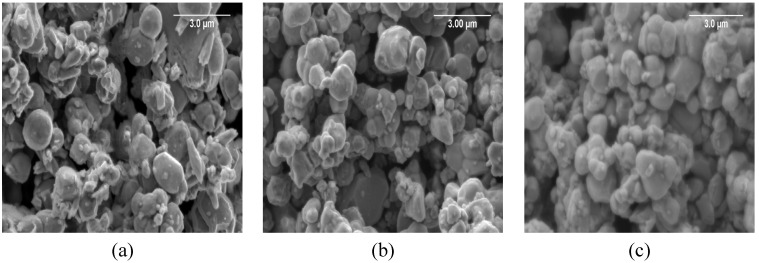
SEM images of the polycrystalline powders: BMW (**a**), BMW-Eu (**b**), and BMW-Nd (**c**).

**Figure 4 materials-14-05996-f004:**
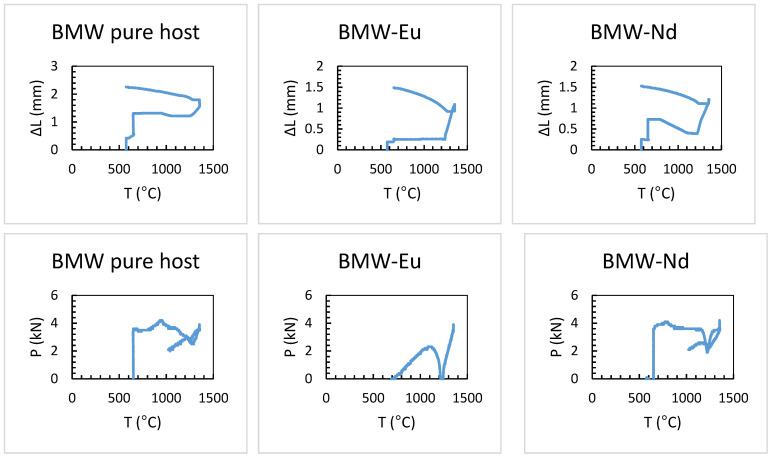
Sintering profiles of double perovskites of Ba_2_MgWO_6_ in function of temperature, where P is pressure and ∆L is shrinkage.

**Figure 5 materials-14-05996-f005:**
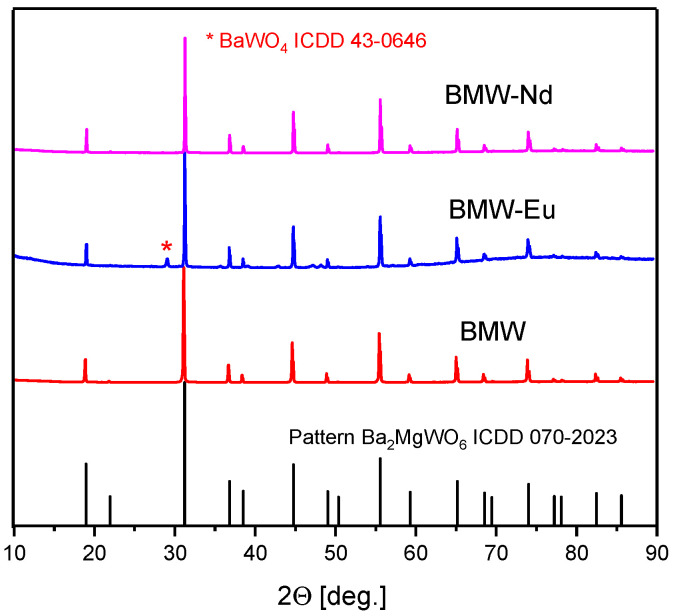
XRD patterns of the ceramics materials: BMW, BMW-Eu, and BMW-Nd.

**Figure 6 materials-14-05996-f006:**
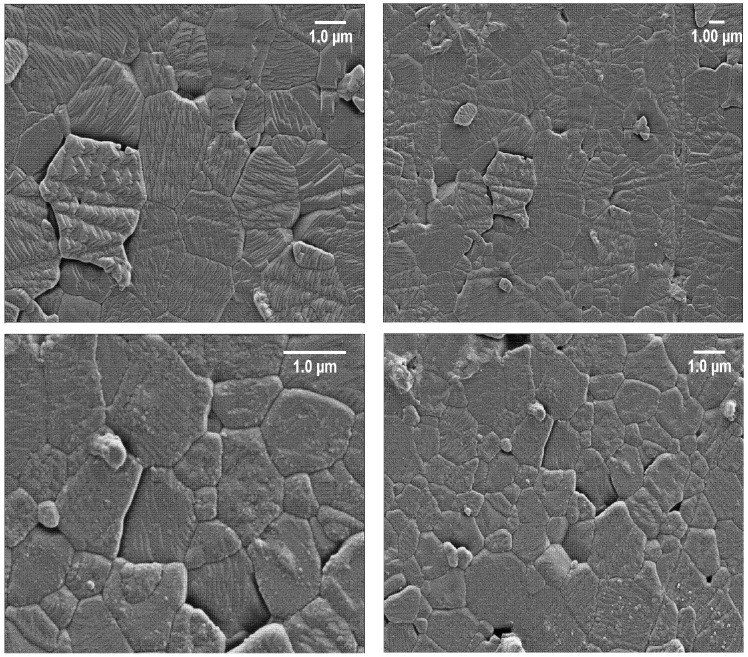
Microstructure of ceramics sintered at 1350 °C by the SPS method: BMW-host (**top row**), BMW-Eu (**center row**), BMW-Nd (**bottom row**). The images on the right and left sides only differ in terms of magnification.

**Figure 7 materials-14-05996-f007:**
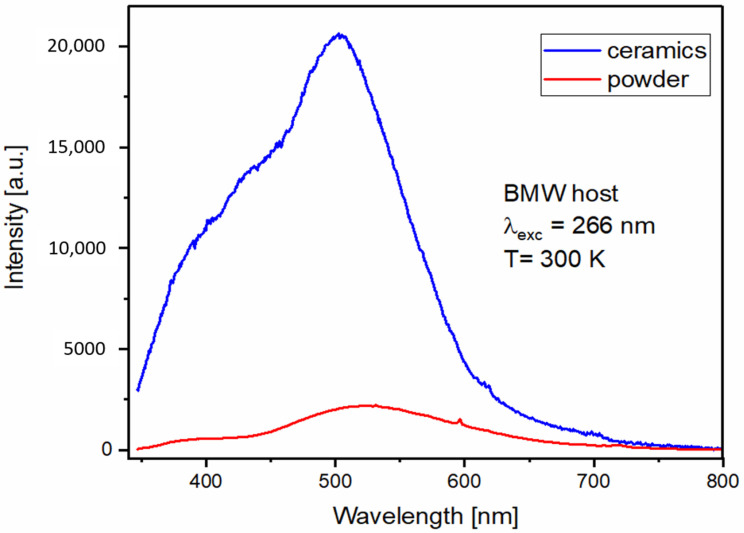
300 K emission spectra of BMW host powder and ceramic under 266 nm excitation.

**Figure 8 materials-14-05996-f008:**
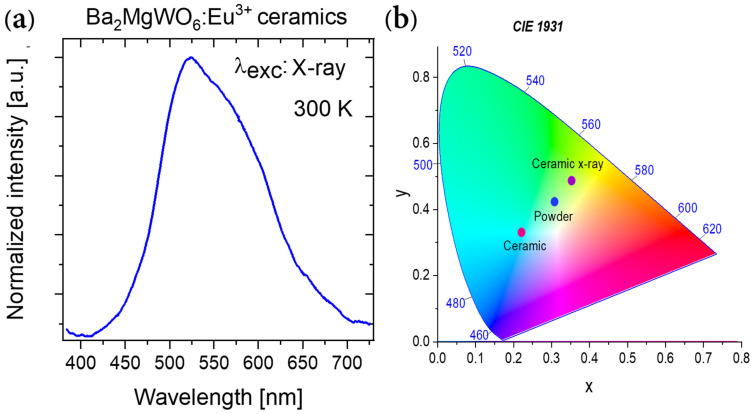
(**a**) 300-K (smoothed) emission spectra of BMW host ceramics under excitation of X-ray source (with 40 kV and 0.7 mA) and (**b**) CIE chromaticity of ceramic and powder BMW host under 266 nm and X-ray excitations.

**Figure 9 materials-14-05996-f009:**
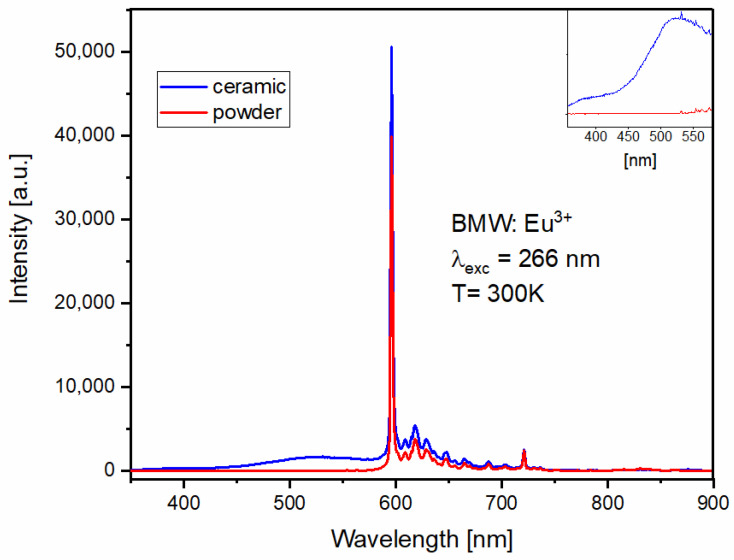
300-K emission spectra of BMW-Eu powder and ceramic under 266-nm excitation.

**Figure 10 materials-14-05996-f010:**
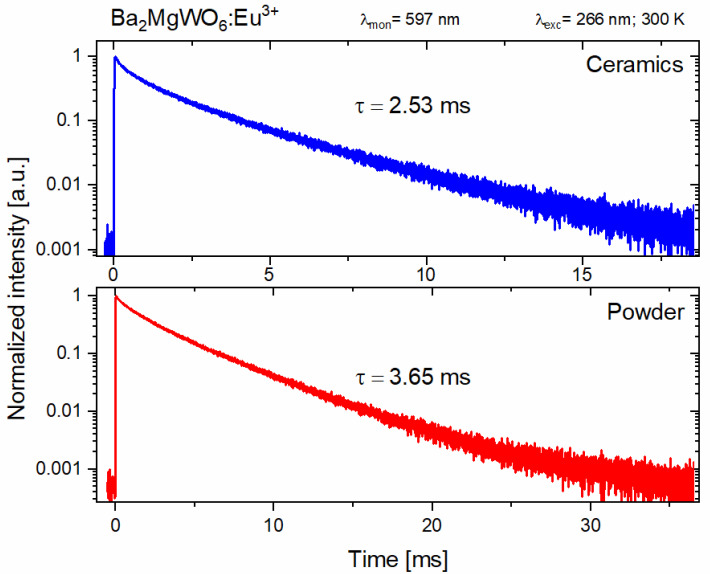
Decay time profiles of the Eu^3+^ emission in BMW host powder and ceramics.

**Figure 11 materials-14-05996-f011:**
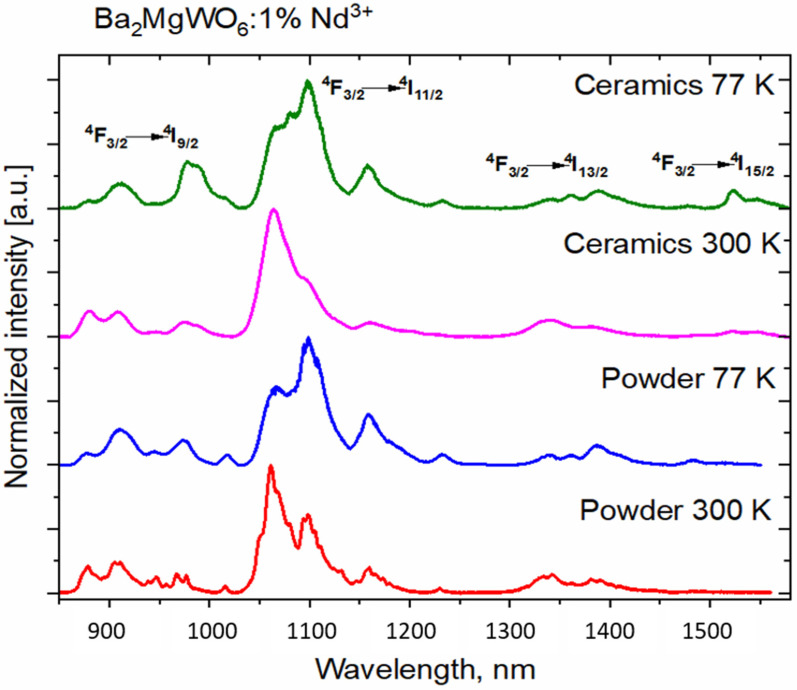
NIR Nd^3+^ emission spectra of BMW-Nd powder and ceramic.

**Figure 12 materials-14-05996-f012:**
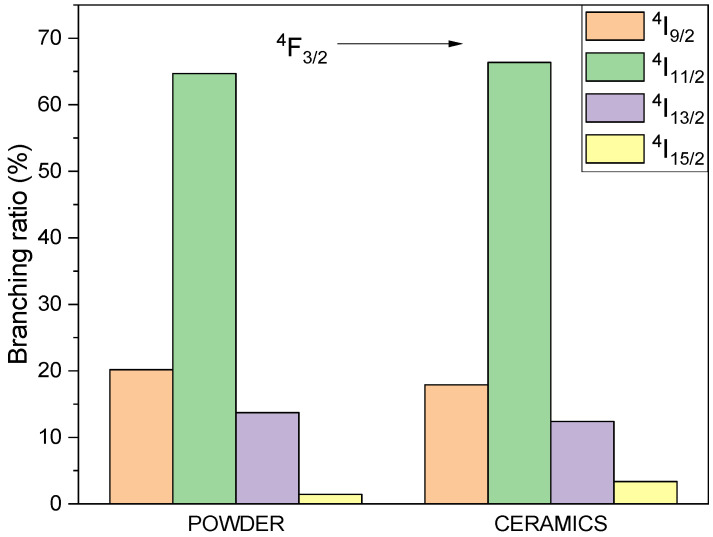
The visualization of the branching ratio between the Nd^3+^ emissions in BMW hosts.

**Table 1 materials-14-05996-t001:** Statement of density parameters of ceramic materials determined by the Archimedes method.

Samples	Apparent Density δ_app_ [g/cm^3^]	Relative Density δ [%]	Open Porosity Π_o_ [%]	Grain Size of Powder [μm]	Grain Size of Ceramic [μm]
BMW host	6.62	91.99	4.29	0.5–2	1.5–3
BMW-Eu	6.09	84.57	6.43	0.5–1.7	0.4–1.4
BMW-Nd	6.60	91.72	4.88	0.7–1.6	1–1.8

**Table 2 materials-14-05996-t002:** The values of the spectroscopic quality parameter for BMW:1% Nd^3+^ samples.

BMW:1% Nd^3+^ Sample/Temperature	Spectroscopic Quality Parameter (SQP)
Powder/300 K	0.65
Ceramics/300 K	1.14

## Data Availability

Data is contained within the article.

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
