# Peer review of "Effect of Ceramic Formation on the Emission of Eu3+ and Nd3+ Ions in Double Perovskites"

_materials, 2021, doi:10.3390/ma14205996_

Round 1

Reviewer 1 Report

The authors have investigated the effect of SPS sintering on the luminescence properties of Ba2MgWO6 and Eu3+ or Nd3+ doped Ba2MgWO6. As the title described the application for scintillators, the authors seemed to try to investigate scintillation properties of the materials. However, the manuscript is not written for the purpose. Each experimental result was scattered and not organized. Furthermore, the explanation to the experimental data was insufficient in many cases. Therefore, it is not accepted for the publication until those deficiencies are fixed. At least, the authors are required to modify the manuscript according to the comments described below.

  1. The doped sample compositions are Ba2Mg0.99Eu0.01WO6 and Ba2Mg0.99Nd0.01WO6. Is there any evidence that Eu3+ and Nd3+ occupy not the W site but the Mg site?

  1. Where are the additional lithium in the samples?

  1. The explanation of Fig. 4 is insufficient. The explanation may be found in their previous paper (ref. 6), however, the detailed explanation is helpful for readers.

  1. There are no captions in Fig. 7. What is difference between the right pictures and the left ones?

  1. At line 266, the authors wrote “some impurities are visible in the form of small grains with darker contrast, especially in the BMW-Eu samples,” however, it is difficult to find where the impurities were.

  1. At line 275, what are the emission of regular and inverse WO6 groups?

  1. At line 306, the referred paper [25] is currently not available. The authors should show details to explain the energy transfer between the regular WO6 and Eu3+ ions.

  1. At line 322, the authors should explain why the longer luminescence decay for the powder sample might be enhanced by the energy transfer from the host.

  1. A variety of luminescence properties were investigated for both powdered and SPSed samples. Some of them showed clear difference between the samples. Why did the differences occur? If this explanation is not provided, the manuscript will be merely a row of experimental results.

  1. Only Fig. 9 provided the scintillation properties. The manuscript did not match the title.

Author Response

Authors of the manuscript Ref. No.: materials-1357696 entitled: “The effect of ceramics formation on the sensitization of rare earth ions for use as scintillators” would like to thanks the Reviewers for the preparation of the review and their comments. These comments are valuable and very helpful for revising and improving our paper. We have studied reviewers’ comments carefully and tried our best to revise our manuscript according to the comments. All changes and new measurements developed in the manuscript are in red.

Reviewer #1: The authors have investigated the effect of SPS sintering on the luminescence properties of Ba2MgWO6 and Eu3+ or Nd3+ doped Ba2MgWO6. As the title described the application for scintillators, the authors seemed to try to investigate scintillation properties of the materials. However, the manuscript is not written for the purpose. Each experimental result was scattered and not organized. Furthermore, the explanation to the experimental data was insufficient in many cases. Therefore, it is not accepted for the publication until those deficiencies are fixed. At least, the authors are required to modify the manuscript according to the comments described below.

ANSWER:

Thank you very much for reviewing our manuscript. We have addressed all of the listed points below. 

  1. The doped sample compositions are Ba2Mg99Eu0.01WO6 and Ba2Mg0.99Nd0.01WO6. Is there any evidence that Eu3+ and Nd3+ occupy not the W site but the Mg site?

ANSWER:

Thanks for this question because we were also tackling this problem and to solve this puzzle the volume of the BMW unit cell was checked according to the concentration of Eu3+. The volume of the BMW unit cell increases with the concentration of Eu3+ which proves that Eu3+ occupies the position of a smaller ion. It was published in [N. Miniajluk, et al., Eu3+ ions in a high symmetrical octahedral site in the Ba2MgWO6 double perovskite, Journal of Alloys and Compounds 802 (2019) 190-195]. Please note also that the W6+ ion is even smaller than that of Mg2+ (see table below) but more importantly, there is too much difference in charge, which would require large charge compensation and ultimately lead to many imperfections in the crystal lattice. The host would create a lot of vacancies to locally compensate the charge.

.

Table. 1. Ionic radius values of individual ions in the host.

Ion in the host

Coordination number

6

8

Ba2+

-

156.0 pm

Mg2+

86.0 pm

-

W6+

74.0 pm

-

Li+

90 pm

Eu3+

108.7 pm

120.6 pm

Nd3+

112.3 pm

124.9 pm

In addition, the emission spectra in 10 K were measured. The obtained results showed that the electro-dipole transitions are very weak compared to the dominant magnetic dipole 5D07F1, which indicates a high symmetry of the site with inversion center, confirming the position of Eu3+ ions in the highest symmetry Oh of Mg site. For these three reasons, we believe that such a situation where Eu3+ replaces with W6+ is highly unlikely.

  1. Where are the additional lithium in the samples?

ANSWER:

Thank you for this question. Li+ ions were chosen as a charge compensator knowing that Mg2+ replace with Eu3+. Without Li+ ions, the crystal lattice had to create an Mg2+ vacancy to compensate for the charge. On the other hand, Eu3+ with Li+ form pairs taking the place of two adjacent Mg2+ ions and thus leave the BMW crystal structure in charge equilibrium. Please note that Li+ is the best choice as a charge compensator since its ionic radius is very close to Mg2+ (see Table 1).

  1. The explanation of Fig. 4 is insufficient. The explanation may be found in their previous paper (ref. 6), however, the detailed explanation is helpful for readers.

ANSWER:

Thank you for the comments. Fig. 4 was completely removed from the manuscript since it was presented already. Instead, a following paragraph has been added:

“In our previous work [6], an analysis of the thermal behavior with the natural sinter-ing method was performed using dilatometry and the TG-DSC analysis in air, which supplement the test results obtained. The measurements were caried out on the BMW host and were described in detail in the previous paper [6] and therefore were not included in this paper. Overall, the dilatometry results show that our BMW powders have excellent compaction ability, which may be related to the low friction and rearrangement forces experienced during compaction. In addition, the sinterability of BMW synthetic powders remains low during natural sintering, which can be explained by the large grain size. The obtained results of TG-DSC indicate that the sintering temperature of this material should not exceed 1400 °C. For this reason, SPS sintering method Has been studied to help lower the densification temperature. In conclusion, Ba2MgWO6 powder with a double perovskite structure can be used to obtain ceramics.”

  1. There are no captions in Fig. 7. What is difference between the right pictures and the left ones?

ANSWER:

Thank you for the comments. Of course, We agree with the reviewer's opinion that the description of Fig. 7 (now it is Figure 6) may be completely unclear for the reader. Therefore a new capture was proposed:

“Figure 6. Microstructure of ceramics sintered at 1350oC by SPS method: BMW-host (top row), BMW-Eu (center row), BMW-Nd (bottom row). The images on the right and left sides only differ in terms of resolution.”

  1. At line 266, the authors wrote “some impurities are visible in the form of small grains with darker contrast, especially in the BMW-Eu samples,” however, it is difficult to find where the impurities were.

ANSWER:

Thank you for the comments. Indeed statement “some impurities are visible in the form of small grains with darker contrast, especially in the BMW-Eu samples,” is not formulated precisely,  therefore additional paragraph was added to the manuscript:

“The dark spaces visible in the SEM images are pores, while the impurities of the structure in the form of an additional phase BaWO4 are visible in the images in the form of heterogeneity - bright and darker areas in the form of streaks are visible. However, it should be emphasized, that the additional phase in the form of BaWO4 is much more visible on the X-ray diffraction pattern in the form of an additional reflex - the SEM images only confirm this. “

  1. At line 275, what are the emission of regular and inverse WO6 groups?

ANSWER:

Thank you for the comments.. The corresponding text in the manuscript has been changed to:

“The maximum of BMW powder emission is at 522 nm and the emission spectrum (see Fig. 7) consists of two bands assigned to the blue-green emission of regular (in the range 350-500 nm) and yellow-green emission (in the range 450-800 nm) of inverse WO6 groups.”

  1. At line 306, the referred paper [25] is currently not available. The authors should show details to explain the energy transfer between the regular WO6 and Eu3+

ANSWER:

Thank you for the comments. I agree with the opinion of the reviewer that the publication is currently not available because it was sent under a different title again to another journal [T. H. Q. Vu, B. Bondzior, D. StefaÅ„ska, N. Miniajluk-GaweÅ‚, M. J. Winiarski, P. J. DereÅ„. „Changing the sensitivity and operating range of the Ba2MgWO6:Eu3+ optical thermometer by applying the distinct synthesis routes”, the article sent to Scientific Reports, in response to reviewers' opinions, DOI: https://doi.org/10.21203/rs.3.rs-512669/v1.].

The following description of the energy transfer between the regular ions WO6 and Eu3+ is provided in the manuscript:

“Bode and Oosterhout [26] assigned blue-green and yellow-green emission bands to the regular and irregular emission of the WO6 group, respectively. In the latter, the tungsten ions, instead of entering their permanent place at BMW, replace the Mg2+ position. It has been shown [25] that with the increase of Eu3+, the emission intensity of WO6 decreases, which confirms that the excitation energy is transferred from the host to the dopant. Based on Dexter's theory, Blasse analyzed the energy transfer from WO6 groups to Eu3+ ions [27]. He argues that the probability of energy transfer through the exchange mechanism is equal to the product of the overlap of the orbital wave functions W6+ and Eu3+ and the overlap of the emission energy of the WO6 groups with the states of absorption of Eu3+ ions. The orbital overlap and hence the energy transfer mechanism is regulated by the angle between the W6+ -O2- - Eu3+ bonds - if the angle is 90º, the orbital overlap is negligible, and the largest is observed for the angle of 180º [9, 25]. In the previous report on BMW: Eu3+ prepared by the mechanical-chemical method [9], it was calculated that the angle between W6+ (regular) -O2- Eu3+ is equal to 180º, however the angle between W6+ (irregular) -O2--Eu3+ is equal to 135º. This is why energy transfer to Eu3+ ions mainly occurs from regular WO6 groups.”

  1. At line 322, the authors should explain why the longer luminescence decay for the powder sample might be enhanced by the energy transfer from the host.

ANSWER:

Thank you for the comments. This sentence should be changed because it is incorrect. Instead, it is written:

“The shorter lifetime of the Eu3+ emission in the BMW ceramic material is as a result of the deformation of the vicinity of the Eu3+ ions due to a large pressure applied to the powdered materials during sintering. Even a slight distortion of the environment will lower the ideal Oh symmetry with the inversion center in which the Eu3+ ions are located, making the observed emission more likely and therefore shortening the lifetime of the excited 5D0 level.”

  1. A variety of luminescence properties were investigated for both powdered and SPSed samples. Some of them showed clear difference between the samples. Why did the differences occur? If this explanation is not provided, the manuscript will be merely a row of experimental results.

ANSWER:

Thank you for the comments. In the summary of chapter d. Spectroscopic properties of BMW double perovskite of the manuscript, the following explanation is provided:

„ The differences in luminescent properties between powdered and SPSed samples are mainly due to three reasons.

The first has already been mentioned in the description of the lifetimes of the emissions. The high pressure applied during the sintering of ceramics causes slight changes in the symmetry of the environment of Eu3+ ions. Even slight changes in the environment reduce the ideal symmetry of the ions observed for the powder material. In the lowered symmetry, the participation of the electric dipole in the transitions increases, and, therefore, the ion emission transitions, either Eu3+ or Nd3+, become more admissible because the selection rules are partially released. This in turn results in a shorter lifetime of the excited level and a higher emission intensity.

The second reason for the differences in spectroscopic properties of the BMW powder and ceramic materials is the influence of the pressure used in sintering ceramics on the formation of more WO6 inversion sites. This can be seen in particular in Figure 9, which shows a much larger share of the emission of irregular groups of WO6, while in the powder material this emission is almost absent (see inset in Fig. 9). This observation is also confirmed by the approximately 6% reduction in the efficiency of transfer of emissions from the host to the dopant for ceramics. As mentioned above, the irregular groups WO6 hardly transfer energy to Eu3+.

Finally, the third effect of changes in the emission properties of ceramic materials results from the reduction in surface area and grain boundaries that occur in powder samples. This reduction primarily affects emission intensity by eliminating light scattering at grain boundaries and improving overall light output.”

  1. Only Fig. 9 provided the scintillation properties. The manuscript did not match the title.

ANSWER:

Thanks a lot for your comments. Unfortunately, in this case, I cannot agree with the reviewer's opinion. Figure 9 (now it is Figure 8) shows the emission spectrum of BMW pure ceramics, excited by using the X-ray radiation, and in fact, this result confirmed our hypothesis that the BMW material can be used as a scintillator. The other structural and spectroscopic results presented complement this hypothesis. The number of figures is dictated by the fact that the structural analysis is presented first, and then the spectroscopic one.

Reviewer 2 Report

The topic of the article is very important and interesting.
But the work does not present data on the effect of ceramics formation on the sensitization of rare earth ions for use as scintillators. 
Remarks:
1.Figure 3.  - There are no required symbols in the figure. It is not clear which samples parts of the figure belong to. The extreme right side of the figure is of poor quality.
2.  Figure 5.  - the drawing style is not very good. There is no explanation of what the value P (kN). It is not clear what value these results represent for the topic of the paper.
3. Figure 7. - There is no explanation that the right and left side of the picture. There are no necessary designations (a, b, c).
4. Figure 8., Figure 10. - There is no explanation of how it was possible to compare the luminescence intensity values of powder and ceramics. How identical were the excitation conditions?
5. Figure 10 - Insert in figure without units along  axis y, the value is not specified along the  axis x.
6.  It is doubtful to estimate the value of equation 5. How specifically  was the integral intensity of the sample with Eu3+ and the host estimated, and in what spectral range?
7. For what purpose was the luminescence spectrum of the neodymium sample measured at 77 K? Why were such measurements not taken for a sample with europium and a "clean" sample?

Author Response

Authors of the manuscript Ref. No.: materials-1357696 entitled: “The effect of ceramics formation on the sensitization of rare earth ions for use as scintillators” would like to thanks the Reviewers for the preparation of the review and their comments. These comments are valuable and very helpful for revising and improving our paper. We have studied reviewers’ comments carefully and tried our best to revise our manuscript according to the comments. All changes and new measurements developed in the manuscript are in red.

Reviewer #2: The topic of the article is very important and interesting.
But the work does not present data on the effect of ceramics formation on the sensitization of rare earth ions for use as scintillators. 

ANSWER:

Thank you very much for reviewing our manuscript. We have addressed all of the listed points below.

  1. Figure 3.  - There are no required symbols in the figure. It is not clear which samples parts of the figure belong to. The extreme right side of the figure is of poor quality.

ANSWER:

Thanks a lot for your suggestions. Of course, I agree with the reviewer's opinion that “it is not clear which samples parts of the figure belong to”. This is only an editorial error of the text. A corrected description of the figure will be added to the manuscript: “Figure 3. SEM images of the polycrystalline powders: BMW (left), BMW-Eu (center) and BMW-Nd (right). “

I agree with the opinion of the reviewer that “The extreme right side of the figure is poor quality”. A poor quality SEM image was obtained because the sample was loaded very much during the measurement, more so than the other samples. This is due to the fact that, during the measurement, the samples were not covered with any layer: neither carbon nor gold. It was decided not to apply the layer because this would significantly extend the measurement  If, of course, there is such a need, we can repeat the taking of the images with a layer applied to the samples. Nevertheless, there is no certainty that the image quality will improve.

  1.  Figure 5.  - the drawing style is not very good. There is no explanation of what the value P (kN). It is not clear what value these results represent for the topic of the paper.

ANSWER:

Thank you for the comment. I agree with the reviewer’s opinion that “the drawing style is not very good”, because the drawing was prepared in Excel program, not in Origin.

To make the paper more clear a paragraph has been added to the main text (figure number changed on 4):

“Figure 4 shows the sintering profiles of double Ba2MgWO6 perovskites. The upper profiles show the shrinkage of the material as a function of temperature, which tells us if the material has been compacted and if a ceramic material has been formed. While the lower profiles show the change in pressure supplied to the system, also depending on the temperature.”

Unfortunately, as can be seen in the figures, these pressure profiles are difficult to compare with each other because there was a problem with insufficient pressure during the process leading to poor pressure control.

  1. Figure 7. - There is no explanation that the right and left side of the picture. There are no necessary designations (a, b, c).

ANSWER:

Thank you for your comments. Of course, I agree with the reviewer's opinion that the description of Fig. 7 (now it is Fig. 6) may be completely unclear for the reader. The top two images are for BMW-host, the middle two are for BMW-Eu and the bottom two are for BMW-Nd. The images on the right and left sides only differ in terms of resolution. And consequently, the Figure caption has been changed, which reads: “Figure 6. Microstructure of ceramics sintered at 1350 °C by SPS method: BMW-host (top row), BMW-Eu (center row), BMW-Nd (bottom row). The images on the right and left sides only differ in terms of resolution.”

  1. Figure 8., Figure 10. - There is no explanation of how it was possible to compare the luminescence intensity values of powder and ceramics. How identical were the excitation conditions?

ANSWER:

Thank you for the comments. The measurement setup was the same. To be more clear a paragraph has been added to the Experimental part of the paper (figure number changed on 7 and 9, respectively):

“To ensure repetition and comparatives of the results a special setup for measuring the emission in the same condition was applied when it comes to the excitation source and its power, the detector, integration time, geometry, etc. As for the illuminated material, the powder was placed in a flat holder with a quartz window with negligible absorbance, while the ceramic sample was measured without a holder in the same plane as the powder. The spot of the illuminating source was small enough so that the illuminated area of the powder and the ceramic were the same.”

  1. Figure 10 - Insert in figure without units along  axis y, the value is not specified along the  axis x.

ANSWER:

Thank you for the comments. We have included a new Figure 10 in the manuscript, corrected according to the reviewer's suggestions. The figure number has also changed from 10 to 9.

  1.  It is doubtful to estimate the value of equation 5. How specifically was the integral intensity of the sample with Eu3+ and the host estimated, and in what spectral range?

ANSWER:

Thank you for the comment. The equation (5) is mainly used to estimate the energy transfer efficiency between dopants and host, based on the integrated emission intensity. The integrated emission intensity is the area under the spectrum in the range of wavelength 390-800 nm. It was calculated using the proper function of the Origin software than equation 5 was applied:

                 Eq. (5)

Where ,  is the integrated intensity of the sample doped with Eu3+ and the host, respectively.

  1. For what purpose was the luminescence spectrum of the neodymium sample measured at 77 K? Why were such measurements not taken for a sample with europium and a "clean" sample?

ANSWER:

Thank you for the comments. The following explanation will be included in the manuscript:

“All samples were measured both at 77 K and room temperature. The reason the results were not presented in their entirety is the following: only Nd samples exhibit a relevant difference in the emission spectra measured at both temperatures. The emission spectra of other samples are virtually identical in both temperatures.”

Round 2

Reviewer 1 Report

The authors have revised their manuscript according to the reviewers’ comments. Some of the replies did not provide accurate explanations. However, the manuscript has been improved enough to be accepted for the publication on materials. 

Author Response

Reviewer 1: The authors have revised their manuscript according to the reviewers’ comments. Some of the replies did not provide accurate explanations. However, the manuscript has been improved enough to be accepted for the publication on materials. 

ANSWER:

Thank you very much for your positive review. We have indeed put in a lot of work to correct the manuscript in line with the comments of the reviewers.

Reviewer 2 Report

The authors did a good job of fixing the technical notes/
However, the main remark has not been eliminated. We expect, as the title suggests, that the article will prove and demonstrate the applicability of the materials for scintillators. However, this did not happen. The description of luminescent processes alone is not enough for this. 
Much attention has been paid to the synthesis of ceramics.
It may be necessary to change the title of the article. Then everything will fall into place.

Author Response

Reviewer 2: The authors did a good job of fixing the technical notes. However, the main remark has not been eliminated. We expect, as the title suggests, that the article will prove and demonstrate the applicability of the materials for scintillators. However, this did not happen. The description of luminescent processes alone is not enough for this. Much attention has been paid to the synthesis of ceramics. It may be necessary to change the title of the article. Then everything will fall into place.

ANSWER:

Thank you very much for your positive review. As suggested by the reviewer, the title of the work was changed. The current title of the work is: “Effect of ceramic formation on the emission of Eu3+ and Nd3+ ions in double perovskites”. The title has been changed in the manuscript.